# SPARC: Multi-view Spatial Transcriptomics Clustering via Prototypical Contrast and Attentional Fusion

## Abstract

Spatial transcriptomics (ST) technologies measure gene expression along with spatial coordinates, enabling integrative analysis of tissue structure and function. However, existing approaches struggle to fully exploit multi-view complementarity and suffer from an objective mismatch between common pre-training tasks and downstream clustering. We propose SPARC (Spatial transcriptomics clustering via Prototypical Contrast and Attentional Fusion), a unified framework with a multi-level alignment strategy. SPARC introduces three key innovations: Feature-level Alignment via a multimodal attention mechanism that dynamically fuses gene expression and tissue morphology; Distribution-level Alignment via adversarial training to bridge modality gaps in the latent space; and Semantic-level Alignment via a prototype-driven contrastive objective that aligns pre-training with clustering by contrasting samples against learnable semantic prototypes rather than instances, mitigating class-collision. Coupled with a multi-branch GNN and multi-task reconstruction (ZINB for counts and image reconstruction), SPARC yields robust, cluster-friendly embeddings and significantly improves spatial domain identification.

## 1 Introduction

The emergence of spatial transcriptomics (ST) has revolutionized our understanding of cellular heterogeneity and tissue architecture by measuring gene expression while preserving spatial context(Stahl et al., 2016). However, this technology also presents unique computational challenges. ST data are inherently multi-view, typically comprising a high-dimensional gene expression count matrix for each spatial location (spot) and a corresponding hematoxylin and eosin (H&E) stained histology image. The core problem in the field is how to learn a unified, low-dimensional embedding that synergistically combines complementary information from gene expression and tissue morphology while remaining robust to the challenges unique to each modality, such as sparsity, noise, and distributional differences.

To address this challenge, the research community has proposed various computational methods. Early models (e.g., stLearn; (Pham et al., 2023)) relied on shallow integration strategies, using image features to smooth gene expression, which can obscure critical local transcriptional signals and lacks deep fusion capabilities. Subsequently, Graph Neural Network-based methods (e.g., SpaGCN; (Hu et al., 2021)), while integrating multi-source information, are limited by a pre-defined, fixed weighting scheme that statically balances spatial distance and morphological similarity, lacking adaptive flexibility in heterogeneous tissue microenvironments. In recent years, more advanced self-supervised contrastive learning methods (e.g., GraphST; (Long et al., 2023)) have made significant progress, but their commonly used "instance discrimination" pre-training objective is in fundamental conflict with the downstream "spatial domain identification" clustering task. This objective aims to treat each spatial spot as a unique class and distinguish it from all others, which is directly contrary to the goal of clustering–to group similar spots together. This issue, known as "class collision" or the "false negative" problem(Li et al., 2021; Caron et al., 2020), leads to a learned representation space that is suboptimal for downstream clustering analysis, thereby limiting further performance improvements.

To systematically address the these limitations, we propose SPARC, a framework designed for deeper, more adaptive, and task-aligned multi-view representation learning. The core of SPARC is an innovative "multi-level alignment" strategy that confronts existing challenges on three levels. First, at the **feature-level**, it employs a **multimodal attention mechanism** to dynamically learn the fusion weights for gene expression and tissue morphology, achieving adaptive feature alignment. Second, at the **distribution-level**, it utilizes an **adversarial regularization** scheme to bridge the distributional gap between different modalities, learning a more robust and modality-invariant unified representation. Finally, and as its core innovation, at the **semantic-level**, SPARC introduces a **prototypical contrastive learning mechanism**. It no longer treats each spatial spot as an individual class but instead aligns its representation with a set of learnable prototypes representing latent semantic clusters (e.g., tissue layers). This design fundamentally resolves the objective inconsistency between pre-training and downstream tasks, ensuring that the learned feature space is semantically better suited for spatial domain identification and thus overcoming a key bottleneck of existing technologies.

## 2 RELATED WORK

The analysis of spatial transcriptomics (ST) data has rapidly evolved, with methods centered on Graph Neural Networks (GNNs), multi-view data fusion, and contrastive learning. Early GNN-based methods like SpaGCN (Hu et al., 2021) construct a single graph from spatial and expression data, but this can dilute modality-specific information by forcing heterogeneous data into a fixed structure. To better integrate gene expression with histology, methods like stLearn (Pham et al., 2023) use image features to smooth expression data, which risks blurring sharp transcriptional boundaries. More recent approaches have adopted self-supervised contrastive learning. For instance, GraphST (Long et al., 2023) uses an instance discrimination objective, which treats every spatial spot as a unique class. However, this creates "false negatives" by pushing apart spots from the same biological tissue layer, a problem known as "class collision" (Li et al., 2021). This objective is fundamentally misaligned with the downstream goal of clustering similar spots together. Our work, SPARC, addresses these limitations by introducing a multi-level alignment strategy. It uses a multi-branch GNN to preserve modality-specific patterns, an attention mechanism for adaptive fusion, and a novel prototypical contrastive learning objective to directly optimize for clustering, thus resolving the "class collision" problem and aligning the pre-training task with the downstream analysis.

### 2.1 NOTATION DEFINITION

In this paper, we denote matrices by bool uppercase letters (e.g., $\mathbf{A}$) and vectors by bool lowercase letters (e.g., $\mathbf{h}$). Unless otherwise specified, all vectors are regarded as **row vectors**. A column vector is represented by the transpose of a row vector, e.g., $\mathbf{h}^T$. We consider a spatial transcriptomics dataset with $N$ spots. The final embedding dimension is $D_{\text{hid2}}$.

## 3 THE PROPOSED SPARC

In this section, we introduce the architecture and learning strategy of the SPARC model. The overall architecture of SPARC is shown in Figure 1, which is composed of six modules: data preprocessing and multi-view graph construction, multi-view endcoder, multi-view attention, adversarial training, prototypical contrastive learning, and multi-task reconstruction. Following we will give a detailed introduction of each module.

### 3.1 DATA PREPROCESSING AND MULTI-VIEW GRAPH CONSTRUCTION

Generally speaking, the orginal spatial transcriptomic data does not contain graph information and has many invalid spots, therefore, we need to preprocess the data and construct the graph in advance. Specifically, the pre-processing and graph construction details for gene expression data and tissue morphology information data are presented as follows:

**Gene Expression Data Processing:** We employ a **hybrid feature selection strategy**, which combines *Highly Variable Genes (HVGs)* and *Spatially Variable Genes (SVGs)*. This approach not only considers the degree of variation in gene expression but also incorporates spatial information,

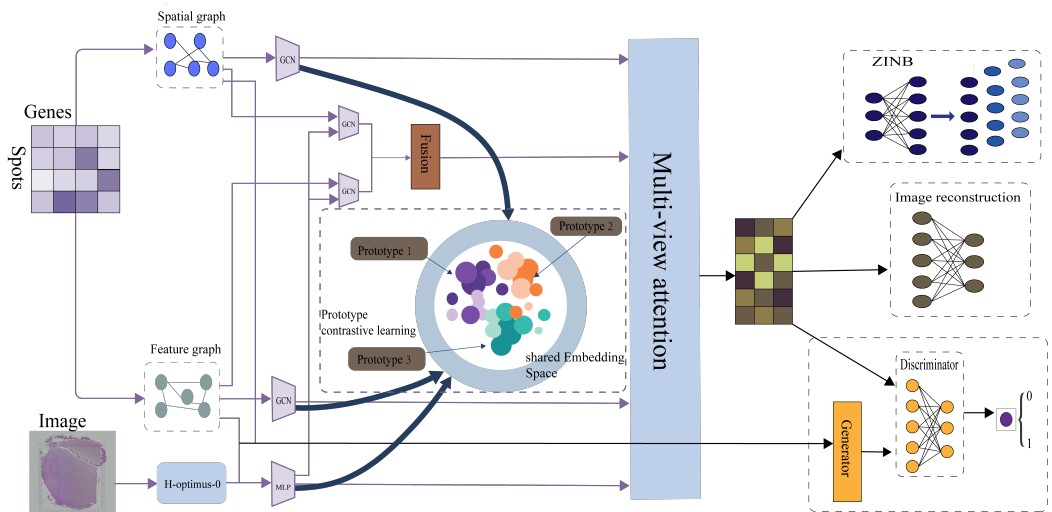

Figure 1: The overall architecture of the SPARC framework.

thereby enabling the selection of a gene subset that is most informative in both biological and spatial dimensions, following standard practices in single-cell/st analysis toolkits such as Scanpy(Wolf et al., 2018).

**Tissue Morphology Information Processing:** For tissue morphology information, we first extract the corresponding image patch for each spot from the hematoxylin and eosin (H&E) stained image. We then utilize a pre-trained vision backbone such as ResNet-50(He et al., 2016) to extract a deep feature vector for each image patch.

**Multi-view Graph Construction:** To capture tissue structure information from different yet complementary perspectives, we construct three primary adjacency matrices and use the processed gene expression and morphological features as node attributes. This multi-view strategy provides a richer source of signals for the subsequent adaptive fusion module, with the design motivation being to allow the model to learn information from multiple "expert" perspectives:

- **Spatial Adjacency Graph** ($A_{\mathbf{spatial}}$)**:** Based on the 2D spatial coordinates of the spots, a spatial neighbor graph is constructed using the k-nearest neighbors (k-NN) algorithm or a fixed radius. This graph captures the physical proximity of spots in the tissue.
- **Feature Adjacency Graph** ($A_{\mathbf{feature}}$)**:** A k-NN graph is constructed based on the similarity of the high-dimensional gene expression vectors between spots. This graph captures transcriptional similarity, connecting spots with similar expression profiles regardless of their spatial location.
- **Morphology Adjacency Graph** ($A_{\mathbf{morphology}}$)**:** Based on the similarity of image feature vectors extracted from histology, a k-NN graph is constructed. This graph captures morphological similarity and connects visually similar points.

### 3.2   MULTI-VIEW ENCODER VIA A MULTI-BRANCH NEURAL NETWORK

SPARC employs a multi-branch neural network to process the different views and graph structures, allowing the model to learn specialized representations from distinct perspectives. This design decouples the initial representation learning process before adaptively fusing the learned embeddings. The architecture consists of two primary gene expression branches and a sophisticated two-stage image feature processing pipeline.

#### 3.2.1   GENE EXPRESSION REPRESENTATION BRANCHES

Two parallel Graph Convolutional Network (GCN) branches(Kipf & Welling, 2017) are utilized to capture complementary information from the gene expression data:

- **Spatial GCN (SGCN):** This branch takes the gene expression features as node attributes and the spatial adjacency graph ($A_{spatial}$) as its structure. It learns a spatially-aware gene expression representation $\mathbf{h}_i^S$, capturing regional patterns of gene expression by aggregating information from physically adjacent spots.

- **Feature GCN (FGCN):** This branch also uses the gene expression features as node attributes but operates on the feature adjacency graph ($A_{feature}$). It learns a transcriptionally-aware representation $\mathbf{h}_i^F$, focusing on the intrinsic structure of the gene expression data by connecting spots with similar transcriptional profiles, regardless of their spatial location.

### 3.2.2 TWO-STAGE IMAGE FEATURE REPRESENTATION

To fully leverage morphological information, SPARC adopts a two-stage process to learn context-aware image representations, which goes beyond a single GCN branch:

- **Stage 1: Initial Image Embedding Extraction.** First, a dedicated feed-forward neural network (FFN) processes the high-dimensional image feature vector for each spot. This network, denoted as IGCN-FFN, learns a compact and informative initial image embedding $\mathbf{h}_i^I$. This step captures the local morphological characteristics of each spot independently of its neighbors.

- **Stage 2: Contextual Enhancement via Graph Propagation.** The initial image embedding $\mathbf{h}_i^I$ is then further refined by propagating it across the established graph structures using a shared-weight GCN, which we term the Image-Context GCN (ICGCN). This generates two context-aware image representations:

  - A spatially-aware image representation, $\mathbf{h}_i^{I-S}$, is obtained by propagating $\mathbf{h}_i^I$ on the spatial adjacency graph ($A_{spatial}$).
  - A transcriptionally-aware image representation, $\mathbf{h}_i^{I-F}$, is obtained by propagating $\mathbf{h}_i^I$ on the feature adjacency graph ($A_{feature}$).

  These two enhanced representations are subsequently combined to form a final context-aware image representation, $\mathbf{h}_i^{I-context}$.

The outputs of these branches—a set of row vectors ($\mathbf{h}_i^S, \mathbf{h}_i^F, \mathbf{h}_i^I, \mathbf{h}_i^{I-context}$) where each vector $\mathbf{h} \in \mathbb{R}^{1 \times D_{hid2}}$—are then fed into the adaptive fusion module.

### 3.3 ADAPTIVE INTEGRATION VIA MULTI-VIEW ATTENTION

To effectively fuse the heterogeneous representations from the multi-branch architecture, SPARC employs a multimodal attention mechanism for adaptive, weighted feature fusion. The final spot representation, row vector $\mathbf{h}_i \in \mathbb{R}^{1 \times D_{hid2}}$, is a weighted sum of the outputs from the multiple branches:

$$\mathbf{h}_i = \alpha_i^S \mathbf{h}_i^S + \alpha_i^F \mathbf{h}_i^F + \alpha_i^I \mathbf{h}_i^I + \alpha_i^{I-context} \mathbf{h}_i^{I-context} \tag{1}$$

where the attention weights $\alpha_i^m$ are learned for each spot via a small neural network, allowing the model to autonomously decide the contribution of each information source (gene expression, morphology, and context) for every spot. This provides flexibility in handling tissue heterogeneity, a key advantage over methods with fixed fusion rules.

### 3.4 ENHANCING EMBEDDING QUALITY VIA ADVERSARIAL TRAINING

SPARC utilizes an adversarial training framework to learn a modality-invariant latent space, addressing the distributional gap between gene expression and tissue morphology data. This is achieved through a minimax game between the main encoder (generator G) and a discriminator network (D). The encoder generates a fused representation $\mathbf{h}_i$, while the discriminator aims to distinguish it from an idealized "real" embedding $\mathbf{h}_i^{real}$, which is constructed from the original multimodal features. The generator is trained to produce embeddings that fool the discriminator, forcing it to learn a latent space that mirrors the structure of the idealized real embeddings. The adversarial loss $\mathcal{L}_{adv}$ and

discriminator loss $\mathcal{L}_D$ are defined as:

$$\mathcal{L}_D = -\frac{1}{N} \sum_{i=1}^{N} [\log D(\mathbf{h}_i^{\text{real}}) + \log(1 - D(\mathbf{h}_i))] \tag{2}$$

$$\mathcal{L}_{\text{adv}} = -\frac{1}{N} \sum_{i=1}^{N} \log D(\mathbf{h}_i) \tag{3}$$

where $D(\cdot)$ is the discriminator's output probability. This process pushes the generator to produce high-quality, robust representations that are statistically indistinguishable regardless of their original modality.

## 3.5 SEMANTIC-AWARE ALIGNMENT VIA PROTOTYPICAL CONTRASTIVE LEARNING

To resolve the "class collision" problem of instance-wise contrastive learning, SPARC introduces a cross-modal prototypical contrastive learning mechanism. Instead of contrasting individual instances, we contrast sample representations against a set of learnable prototypes $\mathbf{C} = \{\mathbf{c}_1, ..., \mathbf{c}_K\}$ that represent latent semantic clusters. This aligns the pre-training objective with the downstream clustering task.

The core idea is a "swapped prediction" task: the gene expression embedding $\mathbf{z}_i^S$ of a spot should predict the same prototype as its image embedding $\mathbf{z}_i^I$. We use the Sinkhorn-Knopp algorithm for soft assignment of embeddings to prototypes. The swapped prediction loss $\mathcal{L}_{X-PCL}$ is:

$$\mathcal{L}_{X-PCL} = -\frac{1}{N} \sum_{i=1}^{N} \left[ \log \frac{\exp(\mathbf{z}_i^S \cdot \mathbf{c}_{\sigma(i,I)}^T / \tau)}{\sum_{k=1}^{K} \exp(\mathbf{z}_i^S \cdot \mathbf{c}_k^T / \tau)} + \log \frac{\exp(\mathbf{z}_i^I \cdot \mathbf{c}_{\sigma(i,S)}^T / \tau)}{\sum_{k=1}^{K} \exp(\mathbf{z}_i^I \cdot \mathbf{c}_k^T / \tau)} \right] \tag{4}$$

where $\mathbf{c}_{\sigma(i,I)}$ is the prototype assigned to spot $i$ based on its image features. This objective forces semantic consistency between modalities for each spot and builds an embedding space with an intrinsic clustering structure. This complements adversarial training, which ensures global distributional alignment, by providing local, semantic-level alignment.

## 3.6 PRESERVING INFORMATION INTEGRITY WITH MULTI-TASK RECONSTRUCTION

To ensure the learned embedding $\mathbf{h}_i$ retains key biological information, SPARC employs multi-task reconstruction as regularization.

- **Gene Expression Reconstruction with ZINB**: We model the raw gene counts using a Zero-Inflated Negative Binomial (ZINB) distribution, which accounts for both technical zeros and biological count variations. The model learns to predict the ZINB parameters ($\pi_{ig}$: dropout probability, $\mu_{ig}$: mean, $\theta_g$: dispersion) from the embedding $\mathbf{h}_i$. The ZINB loss $\mathcal{L}_{\text{ZINB}}$ is the negative log-likelihood of the observed counts, forcing $\mathbf{h}_i$ to preserve information needed to reconstruct the original expression profile.

$$\mathcal{L}_{\text{ZINB}} = -\sum_{i=1}^{N} \sum_{g=1}^{G} \log P(x_{ig} | \pi_{ig}, \mu_{ig}, \theta_g) \tag{5}$$

- **Image Reconstruction**: Similarly, an image decoder reconstructs the histology patch from $\mathbf{h}_i$, minimizing a reconstruction loss $\mathcal{L}_{\text{img}}$ (e.g., MSE). This ensures morphological information is also encoded in the final embedding.

## 3.7 OVERALL LOSS FUNCTION

In summary, the SPARC model is trained end-to-end by optimizing a single, comprehensive loss function that integrates all learning objectives. The overall loss function $\mathcal{L}_{\text{total}}$ is a weighted sum of the reconstruction losses, the adversarial loss, and the prototypical contrastive loss:

$$\mathcal{L}_{\text{total}} = \mathcal{L}_{\text{ZINB}} + \lambda_{\text{img}} \mathcal{L}_{\text{img}} + \lambda_{\text{X-PCL}} \mathcal{L}_{\text{X-PCL}} + \lambda_{\text{adv}} \mathcal{L}_{\text{adv}} \tag{6}$$

where $\lambda_{(\cdot)}$ are tunable hyperparameters that balance the contribution of each loss term.

## 4 EXPERIMENTS AND RESULTS

### 4.1 EXPERIMENTAL SETUP

We evaluate SPARC on six public spatial transcriptomics benchmarks: human dorsolateral prefrontal cortex (DLPFC)(Maynard et al., 2021), human breast cancer (BRCA, 10x)(10x Genomics, 2020), mouse embryo (MOSTA)(Chen et al., 2022), BARISTA(Zhao et al., 2023; Chen et al., 2018), and two mouse olfactory bulb (MOB) datasets(Stahl et al., 2016). We use 12 DLPFC slices with expert annotations (151507, 151508, 151509, 151510, 151669, 151670, 151671, 151672, 151673, 151674, 151675, 151676). The gold-standard labels are cortical layers (L1-L6) and white matter (WM). Our central question is whether SPARC's combination of adaptive fusion, adversarial training, and prototypical contrastive learning delivers superior spatial domain identification compared with existing methods.

BASELINE MODEL SELECTION

EVALUATION METRICS

We report Adjusted Rand Index (ARI)(Hubert & Arabie, 1985), Normalized Mutual Information (NMI)(Vinh et al., 2010), and Silhouette Score(Rousseeuw, 1987).

### 4.2 PERFORMANCE ON SPATIAL DOMAIN IDENTIFICATION

To quantitatively assess spatial domain identification, we compare SPARC against baselines on the 12 DLPFC slices using ARI as the primary metric. SPARC achieves clear gains on most slices.

*Baselines.* We compare against stLearn(Pham et al., 2023), CCST(Li et al., 2022), SpaceFlow(Ren et al., 2022), Scanpy(Wolf et al., 2018), GraphST(Long et al., 2023), stMMR(Zhang et al., 2024), DeepST(Xu et al., 2022), MAFN(Zhu et al., 2024), SpaGCN(Hu et al., 2021), and STAGATE(Dong & Zhang, 2022).

Table 1: Adjusted Rand Index (ARI) comparison on DLPFC (10x Visium) slices. This table reports each method's ARI on 12 DLPFC slices.

| Method | 507 | 508 | 509 | 510 | 669 | 670 | 671 | 672 | 673 | 674 | 675 | 676 | Average |
|---|---|---|---|---|---|---|---|---|---|---|---|---|---|
| stLearn | 0.49 | 0.31 | 0.45 | 0.44 | 0.32 | 0.23 | 0.39 | 0.34 | 0.30 | 0.38 | 0.38 | 0.40 | 0.37 |
| CCST | 0.45 | 0.42 | 0.44 | 0.36 | 0.35 | 0.34 | 0.68 | 0.62 | 0.42 | 0.36 | 0.48 | 0.50 | 0.45 |
| SpaceFlow | 0.43 | 0.30 | 0.44 | 0.43 | 0.23 | 0.25 | 0.38 | 0.43 | 0.46 | 0.24 | 0.38 | 0.34 | 0.36 |
| Scanpy | 0.22 | 0.14 | 0.16 | 0.09 | 0.22 | 0.21 | 0.13 | 0.13 | 0.21 | 0.31 | 0.28 | 0.22 | 0.19 |
| GraphST | 0.44 | 0.49 | 0.52 | 0.44 | 0.56 | 0.47 | 0.61 | 0.63 | 0.63 | 0.61 | 0.54 | 0.59 | 0.54 |
| stMMR | 0.59 | 0.51 | 0.57 | 0.66 | 0.49 | 0.50 | 0.69 | 0.64 | 0.59 | 0.51 | 0.57 | 0.58 | 0.58 |
| DeepST | 0.56 | 0.43 | 0.44 | 0.50 | 0.36 | 0.24 | 0.36 | 0.43 | 0.60 | 0.47 | 0.55 | 0.54 | 0.46 |
| MAFN | 0.69 | 0.53 | 0.72 | 0.61 | 0.60 | 0.48 | 0.83 | 0.77 | 0.51 | 0.49 | 0.50 | 0.54 | 0.61 |
| SpaGCN | 0.64 | 0.46 | 0.56 | 0.53 | 0.39 | 0.36 | 0.63 | 0.78 | 0.61 | 0.61 | 0.55 | 0.58 | 0.56 |
| STAGATE | 0.51 | 0.52 | 0.32 | 0.49 | 0.25 | 0.48 | 0.62 | 0.62 | 0.60 | 0.64 | 0.61 | 0.43 | 0.51 |
| **SPARC** | **0.67** | **0.64** | **0.72** | **0.68** | **0.82** | **0.70** | **0.83** | **0.80** | **0.58** | **0.53** | **0.56** | **0.58** | **0.68** |

These results provide strong empirical evidence for SPARC's design–the synergy of adaptive fusion, adversarial training, and prototypical contrastive learning. Visualization of the learned embeddings further shows that SPARC yields clearer, better-separated clusters across tissue regions (Figure 2).

As shown in Figure 2, on DLPFC 151509 SPARC attains higher boundary fidelity and intra-layer consistency, reducing cross-layer leakage and within-layer fragmentation. It also maintains stable segmentation around gyri/sulci boundaries where morphology changes rapidly, which we attribute to the multi-view adaptive fusion and semantic prototype alignment.

#### 4.2.1 RESULTS ANALYSIS

1. **Superior Overall Performance**: SPARC achieved an average ARI score of 0.67, significantly higher than all competing methods. The closest competitors, MAFN and spCLUE, only achieved average scores of 0.60 and 0.57, respectively, which demonstrates the superiority of SPARC's overall architecture.

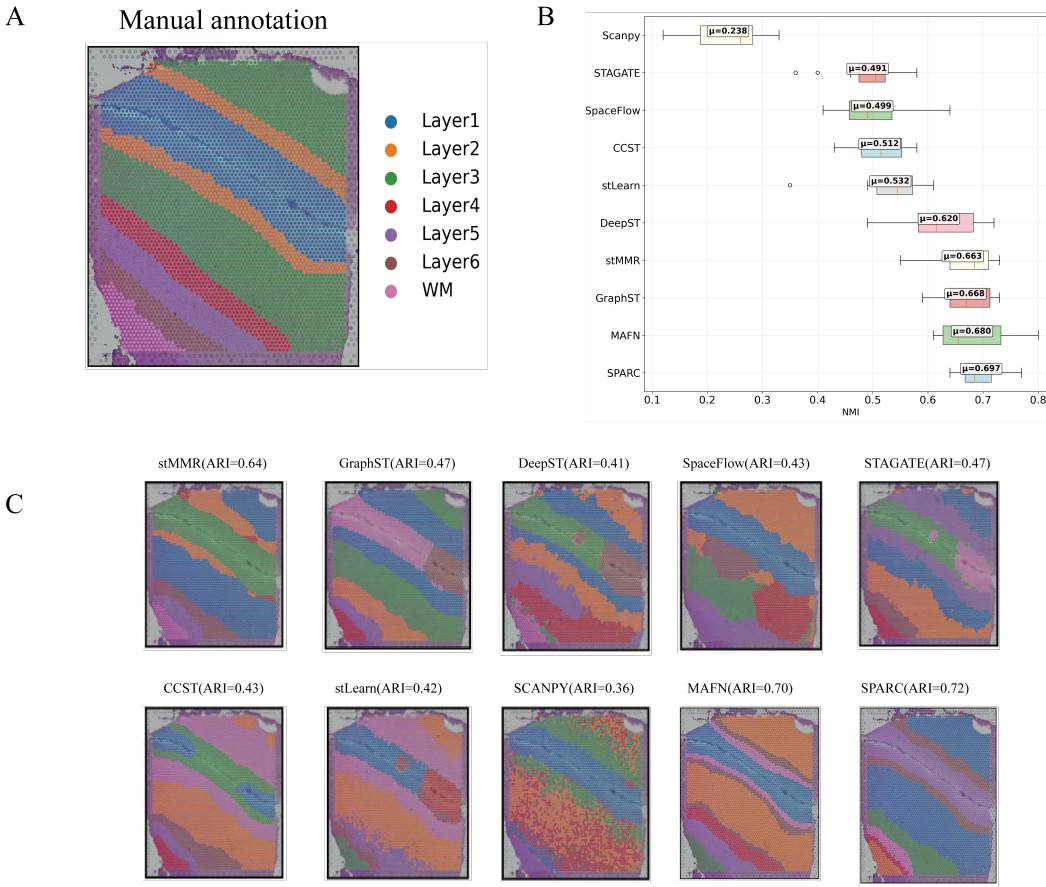

Figure 2: Spatial domain identification on DLPFC slice 151509. (A) Ground-truth annotations (L1-L6, WM) versus SPARC's clustering; (B) visual comparison with 12 baselines (titles show each method's NMI). On this slice, SPARC achieves ARI = 0.72 (see Table 1, column 509), recovering the laminar organization with sharper boundaries at typical interfaces such as WM-L6 and L4-L3. Intra-layer regions are more contiguous with less fragmentation; the few mis-assignments concentrate near inter-layer transition zones and morphology-change boundaries, highlighting both discriminability for laminar structure and robustness around ambiguous edges.

2. **Outperforming the SOTA Model GraphST**: Compared to its most direct competitor, GraphST (average ARI of 0.54), SPARC shows a substantial performance improvement. This result strongly supports our core hypothesis: by directly optimizing for the clustering objective via prototypical contrastive learning, the "class collision" problem inherent in instance-wise contrastive learning can be effectively resolved, leading to a representation space more suitable for downstream clustering tasks.

3. **Robust Performance**: SPARC's advantage is not limited to a few datasets but is demonstrated across multiple datasets, achieving the highest ARI scores on '151510', '151669', '151670', '151671', and '151672'. This indicates that SPARC's adaptive multimodal fusion and multi-level alignment strategy is robust and widely applicable.

Overall, these findings substantiate the synergy among SPARC's core components and corroborate the qualitative improvements observed in the visualizations.

### 4.3 ABLATION STUDY

To dissect the SPARC model and validate the contribution of each of its key innovative modules, we conducted a series of rigorous ablation studies. We systematically removed a key component from

the model (attention fusion, prototypical contrastive learning, or adversarial training), then retrained the model and evaluated its performance.

The experimental results clearly show that removing any of the core components leads to a significant drop in model performance.

- **Removing Attention Fusion**: When the attention module was removed and replaced with simple feature averaging or concatenation, the model's performance declined. This confirms our hypothesis that adaptively weighting different modalities is crucial for handling heterogeneous ST data. A static fusion strategy cannot effectively balance the importance of gene expression and tissue morphology in different regions.

- **Removing Prototypical Contrastive Learning**: When the prototypical contrastive learning module ($\mathcal{L}_{X-PCL}$) was removed, the model's clustering performance dropped substantially. This directly demonstrates the key role of PCL in constructing an embedding space with a clear semantic structure that is conducive to clustering. Without this module, the model loses its ability to optimize directly for the clustering task.

- **Removing Adversarial Training**: Removing the adversarial training module ($\mathcal{L}_{adv}$) also led to a performance decrease, indicating that bridging the distributional gap between modalities is necessary for learning a robust and generalizable multimodal representation. Without adversarial regularization, the model is more susceptible to bias from a single modality.

**Ablation settings.** We compare the full SPARC against the following variants: (1) w/o Attention: replace multimodal attention with simple average of branch features; (2) w/o PCL: remove $\mathcal{L}_{X\text{-}PCL}$; (3) w/o Adv: remove $\mathcal{L}_{adv}$. Optionally, for completeness you may also report: (4) w/o Img recon: remove $\mathcal{L}_{img}$; (5) w/o ZINB: replace ZINB with MSE on normalized counts.

Table 2: Ablation on DLPFC: ARI (mean $\pm$ std) across 12 slices. Replace "TBD" with your results.

| Setting | ARI (mean) |
|---|---|
| SPARC (full) | 0.68 |
| w/o Attention | 0.50 |
| w/o PCL | 0.64 |
| w/o Adv | 0.63 |
| w/o Img recon | 0.65 |
| w/o ZINB | 0.64 |

**Training protocol.** Keep all hyperparameters identical to the full model; change only the component under ablation.

## 5 DISCUSSION

SPARC's superior performance stems from the synergy of its components. At the feature level, multi-branch GNNs learn specialized representations. At the representation level, an attention mechanism performs adaptive fusion. Finally, at the latent space level, adversarial training and prototypical contrastive learning work in concert. Adversarial training achieves global distributional alignment between modalities, while prototypical contrastive learning ensures local, semantic alignment by organizing embeddings around cluster prototypes. This multi-level strategy enables a more robust and comprehensive integration of multimodal data.

Compared to methods like SpaGCN and stLearn, SPARC's adaptive attention fusion is more effective than static or shallow integration schemes. Most importantly, compared to instance-wise contrastive methods like GraphST, SPARC's prototypical contrastive learning directly optimizes for clustering. This resolves the "class collision" problem, where instance discrimination conflicts with the goal of grouping similar spots. This alignment between the pre-training objective and the downstream task is the primary reason for SPARC's state-of-the-art performance, as empirically validated by our results (Table 1).

Despite its success, SPARC's performance relies on hyperparameter tuning, and its current design is for 2D data. Future work could involve integrating SPARC's core innovations, particularly the prototypical contrastive loss, into large-scale, pre-trained foundation models for ST analysis. Extending the framework to 3D spatial omics data is another promising direction.

# 6 CONCLUSION

We introduced SPARC, a multimodal representation learning framework for spatial transcriptomics that integrates a multi-branch GNN, attention fusion, adversarial training, and a novel prototypical contrastive learning objective. By aligning the self-supervised pre-training task with the downstream clustering goal, SPARC resolves the "class collision" problem inherent in instance-wise contrastive methods and achieves state-of-the-art performance in spatial domain identification. Our work demonstrates that a synergistic combination of adaptive fusion and a clustering-oriented learning objective is a powerful strategy for spatial omics analysis. Future work will focus on extending SPARC to 3D datasets and improving its scalability.

# 7 REPRODUCIBILITY AND ETHICS STATEMENT

Experiments were conducted on Ubuntu 22.04 with an NVIDIA RTX 5090 GPU. All dataset acquisition and organization details are provided in Appendix A.3.

We use only publicly available datasets under their original licenses and terms. No personally identifiable or private data are involved; all experiments are computational. All results and conclusions were reviewed by the authors, and LLM usage is disclosed in Appendix "Use of Large Language Models (LLMs)".

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

## A  APPENDIX

### A.1  DATASETS

The LIBD human dorsolateral prefrontal cortex (DLPFC) dataset (http://spatial. libd.org/spatialLIBD)

### A.2  EXTENDED ABLATION RESULTS (PLACEHOLDERS)

This section provides a per-slice ablation template. Replace "TBD" with measured ARI on each slice.

Table 3: Per-slice ARI for ablations on DLPFC.

| Setting | 507 | 508 | 509 | 510 | 669 | 670 | 671 | 672 | 673 | 674 | 675 | 676 | Avg |
|---|---|---|---|---|---|---|---|---|---|---|---|---|---|
| w/o Attention | 0.35 | 0.57 | 0.56 | 0.54 | 0.45 | 0.56 | 0.56 | 0.70 | 0.34 | 0.34 | 0.66 | 0.45 | 0.50 |
| w/o PCL | 0.65 | 0.67 | 0.75 | 0.71 | 0.68 | 0.60 | 0.70 | 0.79 | 0.54 | 0.54 | 0.56 | 0.55 | 0.64 |
| w/o Adv | 0.59 | 0.64 | 0.66 | 0.71 | 0.72 | 0.88 | 0.86 | 0.83 | 0.51 | 0.47 | 0.36 | 0.37 | 0.63 |
| w/o Img recon | 0.67 | 0.68 | 0.71 | 0.58 | 0.64 | 0.64 | 0.67 | 0.78 | 0.62 | 0.60 | 0.60 | 0.58 | 0.65 |
| w/o ZINB | 0.62 | 0.63 | 0.57 | 0.54 | 0.74 | 0.72 | 0.86 | 0.82 | 0.54 | 0.54 | 0.56 | 0.56 | 0.64 |

## A.3 DATASETS AND ACCESS

DLPFC (10x Visium, 12 slices: 151507-151510, 151669-151676). The remaining datasets (BRCA, MOSTA, BARISTA, and two MOB datasets) are all publicly available; acquisition details align with Appendix A.1. After download, organize data under data/DLPFC,BRCA,MOSTA,BARISTA,MOB1,MOB2/ following the original folder structure from the source.

## A.4 USE OF LARGE LANGUAGE MODELS (LLMs)

We used an LLM-based assistant to:

- polish writing for clarity and readability;
- suggest layout/structure refinements (e.g., section headings, figure/table placements);
- check grammar, style consistency, and terminology/notation alignment.

All methods, experiments, analyses, and conclusions were conceived and executed by the authors. All LLM-suggested edits were reviewed and verified by the authors prior to adoption. No private or confidential data were shared with the LLM; only manuscript text and public citations were used.

