# OpenReview forum: "SPARC: Multi-view Spatial Transcriptomics Clustering via Prototypical Contrast and Attentional Fusion"
_ICLR.cc/2026/Conference — ICLR 2026 Conference Withdrawn Submission_

### Official Review · Reviewer_iWBg · 2025-10-30

**Soundness:** 1
**Presentation:** 1
**Contribution:** 2
**Rating:** 0
**Confidence:** 5

**Summary:**

This paper designs a multi-view spatial transcriptomics clustering method via prototypical contrast and attentional fusion. Experiments are conducted on one dataset to verify its effectiveness.

**Strengths:**

N/A

**Weaknesses:**

1. Poor representation, equation formulation, and flowchart in the paper.

2. Lack of clarity in explaining the motivation behind this paper.

3. Use of outdated technology in the paper. The weighted fusion manner, prototype contrastive learning, and adversarial learning are commonly used in previous multi-view or multi-modal learning methods.

4. Limited experiments conducted; more experiments needed for robust conclusions. In recent years, there have been several foundation models in this field. I suggest that the authors compare with them, showing the advantages and disadvantages of the proposed method.

**Questions:**

Please see Weaknesses.

---

### Official Review · Reviewer_fj6M · 2025-11-01

**Soundness:** 3
**Presentation:** 3
**Contribution:** 3
**Rating:** 6
**Confidence:** 4

**Summary:**

This paper proposes a framework for ST clustering that combines multi-view graph neural networks with three alignment strategies: feature-level (multimodal attention), distribution-level (adversarial training), and semantic-level (prototypical contrastive learning). The method constructs three adjacency graphs (spatial, feature-based, morphology-based) and processes them through separate GCN branches before fusing representations via learned attention weights. The key claimed innovation is using prototypical contrastive learning instead of instance-level contrastive learning to avoid class collision problems. Results on various datasets show improvements over baselines.

**Strengths:**

- The motivation for prototypical contrastive learning is well-articulated. The observation that instance discrimination objectives conflict with clustering tasks is valid and the proposed solution of contrasting against learnable prototypes rather than instances is conceptually sound.
The multi-view graph construction (spatial, feature, morphology) is reasonable for capturing complementary information in ST data, and the two-stage image processing (initial embedding + graph propagation) shows careful consideration of how to incorporate histological information.
- Experimental results demonstrate consistent improvements across multiple DLPFC slices with average ARI of 0.68 versus 0.54 for GraphST, which is substantial.
- The ablation study attempts to validate individual components, showing that removing attention, PCL, or adversarial training degrades performance.

**Weaknesses:**

- The prototypical contrastive learning formulation omits how the number of prototypes is selected—a fundamental issue since choosing it equal to the true number of clusters requires ground truth labels, contradicting the unsupervised setting. The projections from the fused embedding to modality-specific embeddings used in the contrastive loss are never defined, the prototype initialization and update mechanism is unclear, and the assignment algorithm is mentioned but not described. Similarly, the adversarial training component never specifies how the real embedding target is constructed or provides the discriminator architecture. The attention mechanism shows weighted fusion but never formalizes how attention weights are computed or what inputs they use. These omissions make the method impossible to reproduce.
- The per-slice ablation results reveal concerning instability: removing adversarial training actually improves performance on some slices compared to the full model, and removing attention achieves comparable results on others. Yet there are no error bars, significance tests, or cross-validation across the tissue slices. The marginal differences between several ablations could easily be within noise levels. The paper relies almost exclusively on one metric in the main table without systematically reporting other clustering metrics.
- The multi-component architecture appears over-engineered without proper justification or ablation. The paper uses separate graph neural network branches for spatial and feature graphs, then adds a two-stage image processing pipeline (feed-forward network followed by graph propagation on both graphs), adversarial training, prototypical contrastive learning, and dual reconstruction losses for expression and image data. However, the ablations don't validate the necessity of this complexity—for example, why is the two-stage image processing needed when gene expression uses single-stage processing? The asymmetric treatment is unexplained. The high variance in ablation results across slices suggests these components may not be universally beneficial but rather overfit to specific dataset characteristics. No comparison against simpler alternatives like concatenated features with standard graph networks is provided to justify the architectural complexity.
- The claim that instance-level contrastive learning is fundamentally flawed for clustering is overstated and not properly validated. While the class collision motivation is conceptually reasonable, the paper doesn't compare against the obvious baseline: instance-level contrastive learning followed by standard clustering algorithms on the learned representations. This comparison is essential to demonstrate that prototypical contrastive learning is actually necessary rather than just marginally beneficial. Methods using instance discrimination have been successfully used for downstream clustering tasks in other domains, so the categorical claim that instance discrimination is not appropriate.

**Questions:**

- How are the learned prototypes and attention weights biologically interpretable—do they correspond to known marker genes or tissue-specific expression patterns?
- How is the number of prototypes selected without access to ground truth labels, and how sensitive is performance to this choice?
- Can authors a direct comparison against instance-level contrastive learning followed by standard clustering to validate that prototypical contrastive learning is necessary?
- What is the computational complexity and runtime compared to baselines, given the multiple graph branches, adversarial training, and iterative prototype assignments?

---

### Official Review · Reviewer_28kR · 2025-11-01

**Soundness:** 1
**Presentation:** 1
**Contribution:** 2
**Rating:** 2
**Confidence:** 4

**Summary:**

This paper proposes SPARC, a spatial clustering method for ST datasets. The method involves three graph constructions (spot coordinates, images, and gene feature) , image feature representations and adversarial training.SPARC also try to preserve each modalities information optimized by ZINB distribution for gene embeddings and element wise MSE comparing image embeddings.

**Strengths:**

1. SPARC compares with 10 prior methods and conducts experiments on popular public datasets.

**Weaknesses:**

1. This paper reads like a sketch for presentation instead of a machine learning paper.
2. The Figure 1 is not legible and low quality. The figure caption is also not informative at all.
3. The reviewer is worried that associating/clustering only the spatial coordinates (not the H&E content and spatial coordinates) with Gene might be a problem that has not much usefulness to study. The reviewer understands that this is an important problem but it also always shows up in the first few samples/tutorial when processing a newly harvested ST dataset but it does not provide much further insights other than some visualizations.
4. The reviewer is worried that in order to show a method works well on such ST dataset, we need a more robust or complete or much more samples of ST slides than the ones being compared in the paper. The reviewer does understand that these are the commonly used samples but concerning these samples and their results won’t draw meaningful conclusions.
5. Spatial clustering or at least the prior methods that this work compares with are mostly taking in only the spatial coordinates (although the reviewer disagrees with such setup) and the gene counts table. Adding in the imaging reconstruction part into the pipeline would make unfair comparison with prior methods.

**Questions:**

1. Thanks for the detailed illustration of your loss function, the reviewer had a hard time understanding the method or the forward pass that you constructed, could the authors please elaborate?
2. Could you briefly provide the motivation for each part of the design of your pipeline?

The current recommendation is a rejection and the reviewer is unlikely to raise the score in the discussion phase.

---

### Official Review · Reviewer_sYJf · 2025-11-03

**Soundness:** 2
**Presentation:** 2
**Contribution:** 2
**Rating:** 4
**Confidence:** 4

**Summary:**

SPARC is a multi-view clustering framework for spatial transcriptomics data. It integrates gene expression and histological information through a three-layer alignment strategy at the feature level, distribution level, and semantic level. Additionally, it introduces prototype contrastive learning to address the inconsistency between pre-training tasks and downstream clustering objectives, significantly enhancing the performance of spatial domain identification.

**Strengths:**

The paper proposes a unified framework named SPARC, which effectively addresses key challenges in spatial transcriptomics analysis through its innovative multi-level alignment strategy. Its core strengths lie in the introduction of a multi-view attention mechanism for adaptive feature fusion, the utilization of adversarial training to bridge distribution gaps between modalities, and most innovatively, the proposal of a prototypical contrastive learning mechanism. By contrasting samples against learnable semantic prototypes, it fundamentally resolves the "class collision" issue between pre-training and downstream clustering tasks, thereby learning a cluster-friendly representation space.

**Weaknesses:**

1. The method fails to demonstrate innovation tailored for specific tasks in spatially resolved transcriptomics data. The adversarial training design objective is outdated, and the definition and construction of relevant data are vague and unclear.
2. The paper does not specify how the prototypes are initialized (e.g., random initialization, K-means initialization), nor does it discuss the impact of different initialization strategies on the stability of the results.
3. The paper lacks a detailed explanation of the hyperparameter tuning process for graph construction and model training, leaving a gap in the analysis of key hyperparameter sensitivity.
4. The model diagram does not clearly illustrate the framework of the method. In particular, the stacked multi-branch GCN encoder makes the overall roadmap cluttered and unclear.
5. Given its complex multi-component architecture, computational cost is a significant practical concern. It remains unclear whether the performance gains justify the high computational overhead compared to more lightweight baseline methods.

**Questions:**

See the Weakness.

---

### Note · Authors · 2025-11-12

I have read and agree with the venue's withdrawal policy on behalf of myself and my co-authors.